# Measurement of Conducted Supraharmonic Emissions: Quasi-Peak Detection and Filter Bandwidth

## Guglielmo Frigo

Swiss Federal Institute of Metrology METAS, Lindenweg 50, 3003 Bern-Wabern, Switzerland;
guglielmo.frigo@metas.ch

**Abstract:** In modern power systems, the integration of renewable energy sources relies on dedicated inverters whose power electronic circuitry switches at high frequencies and causes conducted emissions in the supraharmonic range, i.e., from 9 to 150 kHz. In this regard, the normative framework is still lacking a reference measurement method as well as a set of emission limits and performance requirements. From a metrological point of view, it is important to evaluate whether some of the power quality indices adopted for radiated emissions could be transposed also in this context. In particular, the paper considers a recent algorithm for the identification of supraharmonic components and discusses how its estimates affect the estimation of quasi-peak values. To this end, the paper describes the implementation of a fully digital approach and validates the results by means of an experimental comparison against a traditional quasi-peak detector. The proposed analysis confirms the potential of the considered approach and provides some interesting insights about the reliability of quasi-peak estimation in supraharmonic range.

**Keywords:** supraharmonic; quasi-peak value; spectral analysis; compressive sensing; Taylor–Fourier multifrequency; filter response





## 1. Introduction

In recent years, power systems have been rapidly evolving towards a new energy dispatch paradigm characterized by the ever-increasing integration of renewable energy sources and distributed generation [1,2]. Such resources are connected to the power system by means of dedicated inverters, whose power electronic circuitry switches at high frequency and produces significant pollution in the harmonic (i.e., up to a few kHz) and supraharmonic (i.e., from 9 to 150 kHz) range [3,4].

From a metrological point of view, the measurement of harmonic components is well established, whereas the same does not apply for conducted supraharmonic emissions. Indeed, the current normative framework has not yet selected a reference measurement method for supraharmonic identification and estimation [5,6]. In order to address this need, the recent project SupraEMI [7] has been investigating the rigorous characterization of supraharmonic components in both laboratory and real-world conditions. In view of the complete standardization of supraharmonic emissions in power systems, the project aims at developing rigorous and reproducible measurement procedures, as well as defining plausible emission limits and uncertainty budgets [8,9].

Based on their frequency range, the supraharmonics do not represent a risky disturbance for most of the measurement infrastructure deputed to the monitoring and control of the power system. For instance, phasor measurement units (PMUs) and power quality (PQ) meters are typically characterized by analog front-end and acquisition circuitry that acts as a low-pass filter with a cut-off frequency in the order of a few tens of kHz [10,11]. On the other hand, though, it has been observed that several mass-market electrical appliances suffer from malfunctions induced by supraharmonic interferences. Even the traditional metering and communication infrastructure is affected. In the first case, significant errors

have been noticed in electricity meters when subject to high-frequency emissions [12,13]. In the second one, power line communication (PLC) protocols have shown a serious degradation of the transmitted data throughput in correspondence with concurrent supraharmonic emissions [14,15].

In such a high frequency range, the traditional PQ metrics, e.g., Total Harmonic Distortion (THD) or Spurious Free Dynamic Range (SFDR), may become less significant and capable of capturing the severity level of an interference phenomenon, as they account for the overall distortion level, but do not provide a specific evaluation of the energy in a specific frequency range. Therefore, borrowing the experience of electromagnetic compatibility tests, new performance indices shall be considered. In this sense, a plausible candidate is represented by the quasi-peak (QP) value, which accounts not only for the emission magnitude but also for its repetition rate [16]. Indeed, the experimental practice in modern power systems has shown how power electronic-based emissions have a pulsated nature, depending on the peculiar operating conditions as well as on the network configurations. As a consequence, the computation of the peak or RMS value might vary as a function of the considered observation interval.

Based on these considerations, a reference method for the measurement of supraharmonic components shall cope with diverse and challenging constraints. On one side, it will include a remarkable frequency resolution (i.e., lower than 1 kHz) to discriminate close-by components. On the other side, it will include an accurate time-domain reconstruction for the assessment of emission levels in terms of peak and QP values. Moreover, it will have reduced computational complexity to allow for online measurements (compatible with the processing power and memory resources of currently used instrumentation).

To this end, the recent literature has developed several methods for the precise identification and accurate estimation of supraharmonic components [17], as further discussed in the following section. In this context, the present paper considers a promising candidate based on the joint application of compressive sensing (CS) theory and Taylor–Fourier multifrequency (TFM) models. The theoretical principles as well as a preliminary characterization via numerical simulations are provided in [18]. In this paper, instead, the CS-TFM is employed to derive the QP values associated with the identified supraharmonic emissions. For this analysis, we included in the algorithm a new routine for the digital implementation of a QP detector and we discuss how the uncertainty of CS-TFM estimates propagates to the final result. An experimental validation against a commercial device allows for evaluating the accuracy of the proposed method, as well as its consistency with the traditional approach used in EMC compliance tests.

The paper is organized as follows. Section 2 provides a review of the most promising supraharmonic measurement methods and of the current normative framework. Section 3 describes the proposed method, focusing particularly on the identification of the spectral components and on the digital implementation of the quasi-peak receiver. Section 4 introduces the measurement setup and discuss the results of two measurement campaigns for the experimental validation of the proposed method. Finally, Section 5 provides some closing remarks and outlines the future steps of the research activity.

## 2. Literature Review

### 2.1. Normative Framework

The IEC Std 61000-4-30 [19] includes an informative annex with four alternative methods for harmonic measurements in the 2–150 kHz range. It is worth noticing, though, that none of these methods is recommended or considered as a reference: in the presence of non-constant time patterns of the harmonic component, it is likely that the four methods provide inconsistent results.

In more detail, two methods descend from the well-known CISPR-16 procedure [20] for radiated emissions in the 9–150 kHz range. However, the CISPR-16 procedure is intended for laboratory testing and cannot be easily adapted to a power system scenario: the long measurement time and the high computational requirements are incompatible with online applications [21].

The third method simply extends the Fourier transform algorithm introduced in the IEC Std 61000-4-7 [22]. Once more, the bottleneck is represented by the excessive computational requirements [23]. Finally, the fourth method applies a filter bank to break the 9–150 kHz range into 2 kHz segments and then analyzes each segment separately. Despite allowing for a parallelized solution, this method cannot be implemented in a gap-less fashion and thus neglects the time between consecutive measurements [24].

### 2.2. Supraharmonic Estimation

In recent years, several approaches have been proposed for the identification and estimation of supraharmonic components, as thoroughly summarized in [17].

In harmonic analysis, the vast majority of the estimation approaches consist of customized applications of the Discrete Fourier Transform (DFT), whose resolution and accuracy is improved by means of proper windowing, interpolation and Taylor expansion routines. However, the DFT relies on two main assumptions, namely periodicity and stationarity, which are hardly met in the supraharmonic context. In this sense, a preliminary solution would consist in minimizing the considered observation interval length, but this would result in higher leakage effects and poorer resolution. As a consequence, alternative approaches have been developed and can be indicatively classified into three main categories: demodulation and sub-sampling, wavelet decomposition and CS-based approaches.

In the first case, the input signal is originally acquired with a sampling rate in the order of 1 MHz, in order to satisfy the Shannon–Nyquist criterion and avoid any aliasing effect. Then, the cascade of an analog demodulator and a filter bank splits the input signal into ten partially overlapped bandwidths of 15 kHz. Each demodulated output is then re-acquired at a lower sampling rate (i.e., few tens of kHz) and observation intervals of 5 ms are processed via a DFT-based spectral identification method. Once the frequency and magnitude of the supraharmonic components are identified, a correction stage accounts for the non-ideal behavior of the demodulation and filtering stage [25].

In the second case, the Wavelet Packet Decomposition (WPD) [26], already successfully employed in harmonic analysis, is applied to the supraharmonic range [27]. By means of a zero-crossing detection method, an observation interval corresponding to exactly ten cycles of the fundamental frequency is acquired and iteratively decomposed using the WPD until achieving a frequency resolution of 200 Hz, i.e., compatible with IEC 61000-4-7 and CISPR-16 methods. Thanks to the synchronous sampling configuration, the leakage effects are negligible and no further filtering or processing is required.

Finally, in the third case, the CS theory is exploited to enforce the sparsity assumption and thus obtain a frequency resolution of 200 Hz even with an observation interval of only 0.5 ms [28]. As in many similar CS-based applications, the most challenging aspect is represented by the support recovery stage, whose precision and robustness depend on the adopted dictionary [29–31].

### 2.3. Quasi-Peak Computation

The CISPR-16 specifications were originally conceived for radiated emissions and required the quantification of the QP value as a measure of the interference potential. In this way, pulsated emissions were evaluated as a function of their repetition frequency: pulses with high repetition frequency are expected to be more annoying and harmful and thus should be associated with a higher QP value. On the other hand, it is worth noticing that the QP value is not a linear function of the repetition frequency and thus cannot be directly derived from other envelope measurements, e.g., the peak or the RMS value [32].

Indeed, QP detectors can be also implemented in a purely digital fashion by means of a Short-Time Fourier Transform (STFT) and an Infinite Impulse Response (IIR) filter cascade [33]. Though computationally demanding, similar techniques allow for replicating the exact non-linear behavior of the analog circuit. A more recent and improved implementation exploits a Phase Locked Loop (PLL) technique to expedite the search for the supraharmonic components and thus reduce the overall computational requirements of the STFT stage [34].

## 3. Proposed Method

### 3.1. Performance Target

First of all, it is worth clarifying the performance target of the proposed method in terms of estimation accuracy and computational complexity. The research activity aims at defining a new measurement method for supraharmonic conducted emissions capable of coping with two conflicting requirements. In more detail, the method shall not be limited to stationary laboratory emissions, but shall be able to capture and accurately estimate pulsating emissions as the ones we expect to face in real-world scenarios. On the other hand, the method shall be characterized by a reduced computational and data burden, in order to guarantee a sort of online estimation. Indeed, despite being conceived for standardization purposes, the method is expected to overcome the main limitations of the CISPR-16 method, which remains the most accurate solution for the definition of the reference values.

In this regard, it is important to observe that this paper focuses on the algorithmic details of the measurement method, rather than on its actual hardware implementation. In order to optimize the algorithmic part, the SupraEMI research plan has defined a performance target of 1% maximum uncertainty in stationary conditions and a frequency resolution of 200 Hz [35].

### 3.2. System Configuration

The proposed method has been implemented within the PMU calibration infrastructure developed at the Swiss Federal Institute of Metrology (METAS) [36].

The hardware architecture is shown in Figure 1 and consists of four main elements. The first one is the time reference of the PMU calibrator. On one side, a White Rabbit clock (Seven Solutions, Granada, Spain) is directly linked to the Coordinated Universal Time, namely UTC-CH, and provides a stable pulse-per-second (PPS) trigger signal and a 10 MHz time-base whose deviation from UTC-CH has been quantified to be less than 1 ns. On the other side, a GPS receiver (Meinberg Gmbh, Bad Pyrmont, Germany) allows each measurement to be assigned a timestamp that is traceable to the absolute time. The two following elements are incorporated within an NI PXI 1042Q (National Instruments, Austin, TX, USA). In more detail, the second one is the NI PXI-6682 synchronization board that receives the time reference signals and propagates them to the other boards. By overriding the internal time reference with the outputs of the White Rabbit clock, it is possible to guarantee the higher accuracy and stability of any time signal within the PXI, e.g., the activation triggers as well as the sampling rates of the other boards. The third element is the the NI PXI-6289 data acquisition board. In single-channel mode, the digital-to-analog converter (DAC) is characterized by an output range of $\pm 10$ V, an update rate of 2 MHz and a resolution of 16 bits. Conversely, the analog-to-digital converter (ADC) is characterized by an input range of $\pm 10$ V, a sampling rate of 500 kHz and a resolution of 18 bits. The DAC operates in continuous mode and reproduces the test waveforms that are user-defined and stored in the memory of the PXI. The ADC operates in single mode and acquires the test waveform for an overall duration that ranges from a few ms up to 10 s, depending on the specific test conditions and requirements. Both DAC and ADC are triggered by the same PPS trigger and derive their sampling rate from the same 10 MHz time-base in order to guarantee exact synchronization between generation and acquisition processes. In this sense, the synchronization uncertainty has been quantified as 4.6 µrad, which corresponds

to nearly 15 ns at 50 Hz, i.e., the nominal system rate of power systems [36]. Finally, the fourth element consists of a voltage and current amplifier, CMS 256 (Omicron Electronics Gmbh, Klaus, Austria), and a set of calibrated instrument transformers (i.e., passive voltage dividers and current shunts). The first one increases the generation output range up to 300 V and 32 A, in line with the expected levels in a distribution network scenario, whereas the second ones scales down the test waveforms into a range that is compatible with the ADC analog front-end.

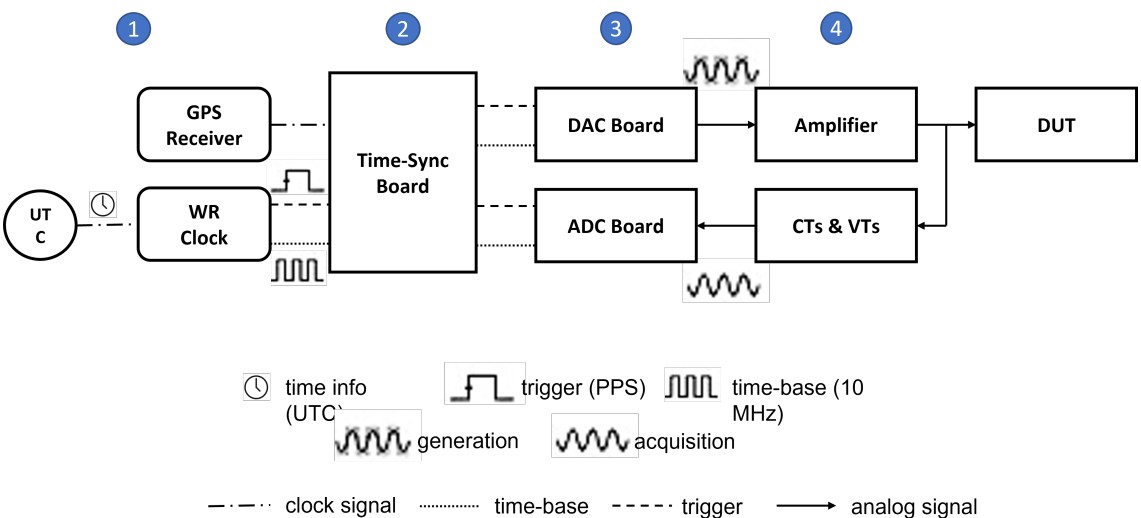

**Figure 1.** Hardware architecture of the METAS PMU calibrator, which serves as a high-precision waveform synthesizer and digitizer in this project.

The software architecture consists of two main elements. The synchronization, generation and acquisition processes are handled in Labview 2021 (National Instruments, Austin, TX, USA). The acquired waveforms are stored in the industrial controller and then processed via a routine in Matlab 2021 (Mathworks, Natick, MA, USA). In view of an online realization, the Matlab routine has been integrated within the Labview code. However, the optimization of the coding structure as well as the minimization of the computation time is beyond the scope of the present paper.

### 3.3. Proposed Method

The proposed method adopts a sampling rate $F_s$ equal to 500 kHz, i.e., sufficiently large to cover the entire supraharmonic range. Given the controlled conditions of the synthesizer and digitizer system, it is reasonable to assume that aliasing effects are negligible or covered by spectral leakage and measurement noise. The acquired waveform is divided into partially overlapped segments of 0.5 ms. The entity of the overlap depends on the selected reporting rate $F_r$, e.g., a reporting rate of 10 kHz corresponds to consecutive segments shifted by 0.1 ms, or equivalently 50 samples.

Differently from other CS-based approaches that consider Multiple Measurement Vectors (MMVs), the proposed method processes each segment separately. Indeed, the proposed method has been designed in order to detect even rapidly pulsating components and the smoothing effect inherent in any MMV approach risks to smooth out too rapid or sporadic emissions. In this regard, it is also worth noticing that the proposed method refers its estimates to the midpoint of the considered segment.

The proposed method consists of four main stages:

- high-pass filtering for the removal of the fundamental component;
- identification of the emission frequencies via CS-based support recovery;
- parameter estimation and time profile reconstruction via TFM decomposition;
- definition of the corresponding QP values via the digital equivalent of a QP detector.

### 3.3.1. High-Pass Filter

In the first stage, an Infinite Impulse Response (IIR) filter is adopted. To this end, the 3 dB cut-off frequency is set equal to 1.5 kHz in order to guarantee a pass-band that includes entirely the supraharmonic range. In this context, Figure 2 shows the frequency response of the high-pass filter. In more detail, the upper plot compares the magnitude and phase response in the supraharmonic range: it is interesting to observe how the filter does not introduce any amplification or diminution in the spectral components as its normalized magnitude response is equal to the ideal value. Conversely, the phase response exhibits an exponentially decaying trend, as reasonably expected in an IIR filter structure. Nevertheless, this aspect does not introduce major concerns as the estimation of emission phases is not considered mandatory in this stage of the research. Moreover, this drawback can be easily solved at the expense of a slightly higher computational burden: either by applying the filter in both forward and backward directions (i.e., zero-phase digital filter), or by applying an a posteriori phase correction based on a look-up table of the filter phase response, as done in [37].

Similar considerations hold for the lower plot, where the filter group delay is expressed in terms of samples. The log-log plot exhibits a linear trend, which corresponds to an exponentially decaying group delay. This aspect might become controversial in the presence of transient events that would trigger slowly dampened oscillations in the filtered signal. Nevertheless, the reduced length of the considered segments allows for minimizing this effect and rapidly stabilizing the filter output.

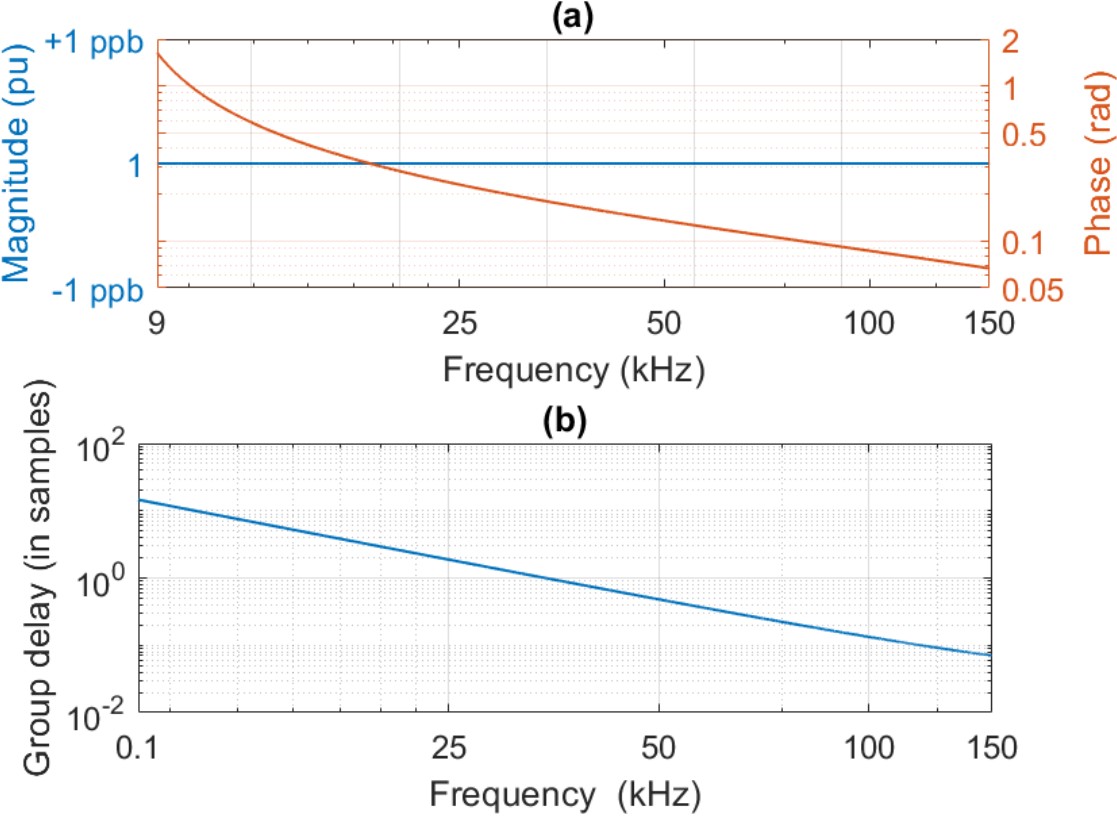

**Figure 2.** (**a**) Magnitude and phase of the filter frequency response in blue and red, respectively. (**b**) Filter group delay as function of input signal frequency.

As a further proof of the effective filter design, Figure 3 analyzes the characteristics of the filter output when supplied with a 50 Hz fundamental component corrupted by a disturbance due to an electronic switch at 25 kHz (further details in Section 4). In more detail, the upper graph shows the magnitude of the filter output DFT: the contributions

related to the fundamental and the lower-order harmonics up to a few kHz have been nullified, whereas the higher portion of the spectral content has been maintained. The lower graph, instead, presents the coherence as deduced by the cross-spectrum of the filter input and output signal: it is interesting to observe how the coherence is maximal in correspondence of the supraharmonic emissions, whereas it rapidly decreases to nearly 0.5 in correspondence of noisy portions. It is thus reasonable to say that the adopted IIR filter allows for removing any leakage contribution from the fundamental component, while preserving the informative content related to the supraharmonic emissions.

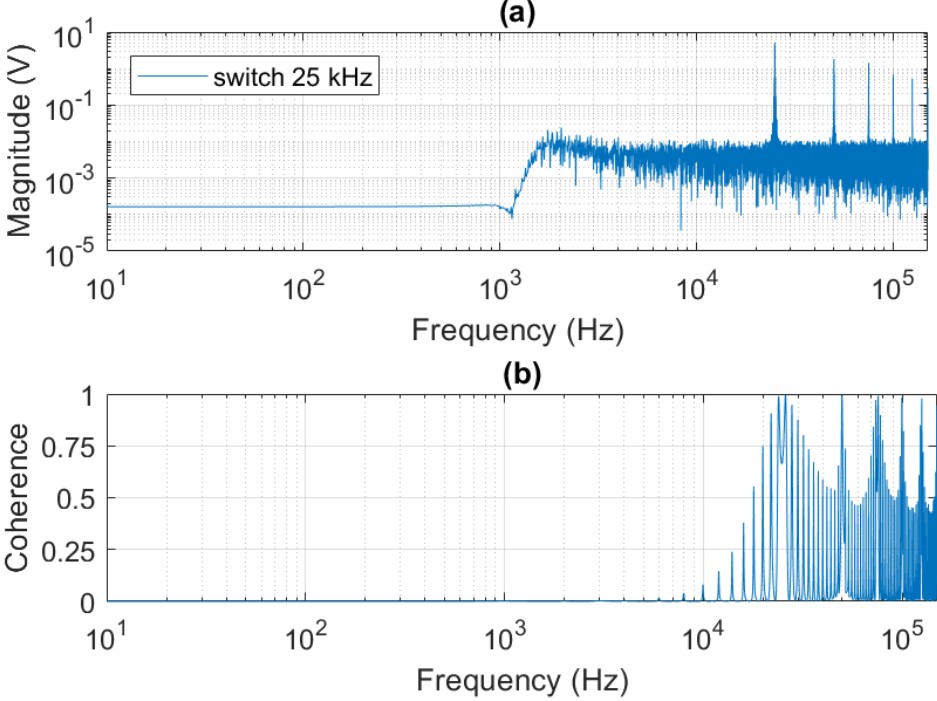

**Figure 3.** (**a**) DFT magnitude of the filter output when supplied with a signal corrupted by a 25 kHz switching event. (**b**) Magnitude-squared coherence between filter input and output.

### 3.3.2. Support Recovery

In the second stage, the signal spectral support is recovered, i.e., the central or dominant frequencies of the most significant emissions are identified. First, the DFT of the considered segment is computed. It is worth underlining that the DFT is defined over a finite grid whose length $N_w$ and resolution $\Delta f_w = F_s / N_w$ are equal to 250 and 2 kHz, respectively. In order to increase the grid granularity, without need of enlarging the segment duration, an overcomplete dictionary **D** is employed [29]: each dictionary column is a Dirichlet kernel defined over a super-resolved frequency grid whose length $N_h = N_w \cdot 13$ and resolution $\Delta f_h = F_s / N_h$ are equal to 3250 and 153.85 Hz, respectively. The dictionary accounts for spectral leakage effects due to the non-synchronous sampling conditions and allows for maximizing the probability of detecting the correct emission frequencies. As a support recovery algorithm, a greedy approach is adopted, namely an Orthogonal Matching Pursuit (OMP) routine [38,39]. As with any other greedy approach, the OMP iteratively selects the frequency that maximizes the projection of the signal residual over the dictionary, and needs a set of stop criteria. In this context, two conditions cause the support recovery routine to stop: either the residual energy is equal to 0.5% of the original signal energy, or the support cardinality (i.e., the number of identified frequencies) is larger than 10. Both these conditions are customizable based on the expected operating conditions [40].

In this context, Figure 4 demonstrates the remarkable reliability of the support recovery stage. In the left plot, the dependence on the noise level is assessed. For this analysis, the test

waveform consists of a fundamental tone and a single supraharmonic component, whose frequencies are set equal to 50.01 and 9920 Hz, respectively, whereas the amplitudes are equal to 230 and 23 V, respectively. It important to observe that the selected frequencies do not belong to the super-resolved grid, but they are nearly placed in the midpoint between two grid bins, i.e., a sort of worst-case condition with maximum leakage and scalloping loss. The test waveform has a duration of 2 s, and the reporting rate is set equal to 50 kHz, thus resulting in an overall set of nearly 100 thousand estimates. For each estimate, the support recovery is considered successful if the identified frequency lies within 200 Hz of the true one (i.e., if the closest bin is selected). The waveform is corrupted by purely additive and uncorrelated white Gaussian noise in order to reproduce different Signal-to-Noise ratio (SNR) levels, ranging from 0 to 40 dB with a step of 5 dB. As shown in Figure 4a, the success probability is nearly optimal at 10 dB (i.e., 97%) and achieves 100% for any SNR larger than 15 dB.

Given an SNR of 40 dB, a second supraharmonic component is added to the test waveform in order to evaluate the support recovery reliability in the presence of close-by components. For this analysis, the frequency separation and the amplitude ratio of the two components are varied: the frequency separation is varied between 2 and 8 kHz, whereas the amplitude of the second component (as normalized by the first one) is set equal to 1, 0.5 and 0.1 pu. As shown in Figure 4b, a minimum frequency separation of 4 kHz guarantees the exact recovery of the spectral support even in the presence of significant mutual distortion (1 pu) or of component amplitudes that are comparable with the noise level (0.1 pu).

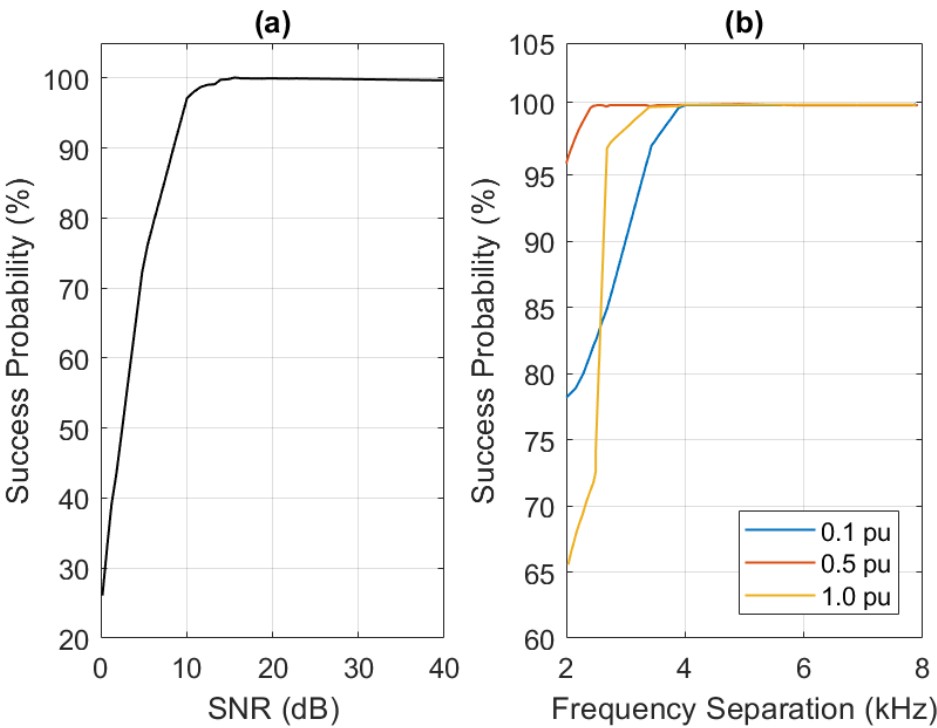

**Figure 4.** (**a**) Support recovery success probability in single-tone conditions as function of SNR. (**b**) Support recovery success probability in multi-tone conditions (with different magnitude ratios) as function of frequency separation [18].

### 3.3.3. Taylor–Fourier Expansion

In the third stage, a TFM model is designed around the frequencies selected in the previous stage. The advantage of applying the TFM technique is two-fold. For each component of the spectral support, the TFM model can be interpreted as a filter bank: a flat differentiator filter around the frequency of interest, and a set of notch filters around the other components (in order to minimize mutual interference). On the other hand, the TFM

expansion exploits the higher-order derivative terms to improve the estimation accuracy of the component parameters (namely amplitude, initial phase, frequency and rate of change of frequency) as well as to reconstruct precisely the component time profile even in the presence of time-varying conditions.

In this sense, Figure 5 provides two examples of TFM filter banks. In the upper plot, a TFM model of a single component is represented as a function of the frequency deviation from the identified component frequency in a single-tone case. The red dashed line represents the -3 dB threshold level. It is interesting to notice how the differentiator filter has a quite large pass bandwidth, thus allowing also non-negligible errors in the support recovering stage. In the lower plot, instead, a detail of the TFM model relative to the multi-tone case in Figure 3b is depicted. More precisely, the plot refers to the supraharmonic component at 25 kHz. It is interesting to observe how the filter response is nearly equal to 0 dB in correspondence of the frequency of interest, whereas it minimizes the interference coming from its harmonic terms, namely at 50 and 75 kHz.

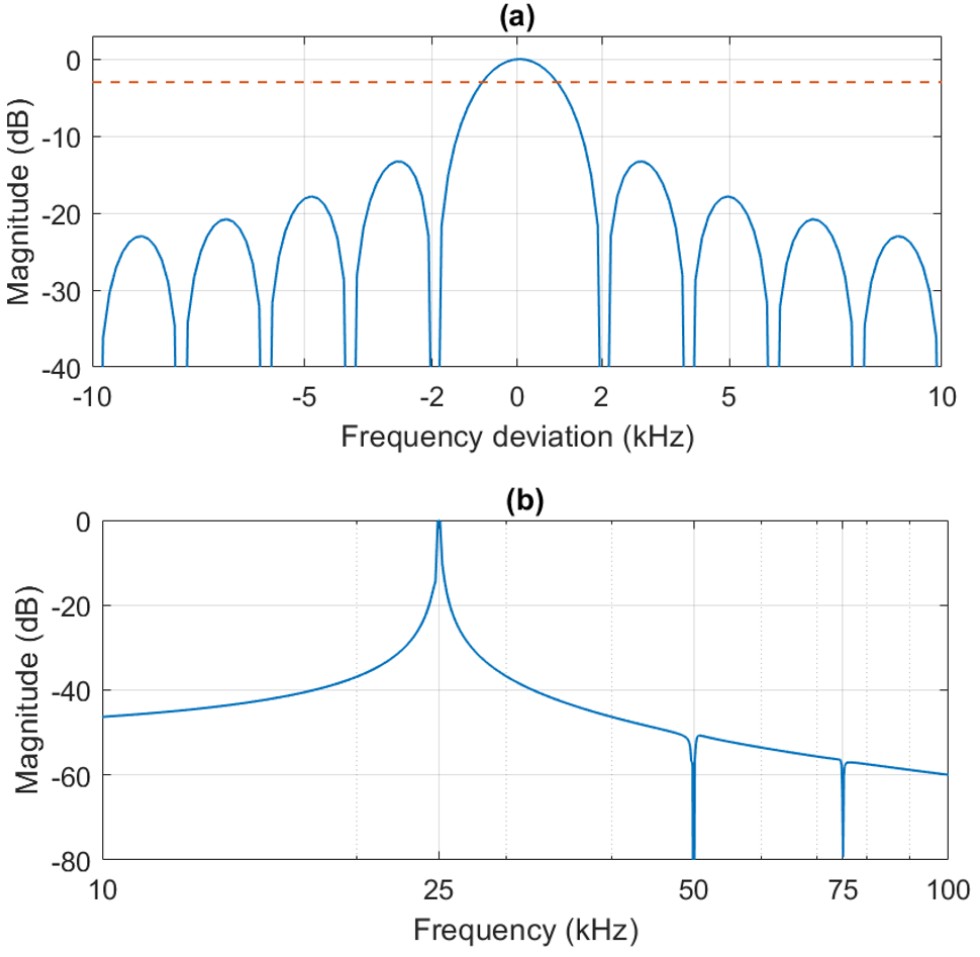

**Figure 5.** TFM filter bank representation: single-tone (**a**), multi-tone (**b**).

The numerical validation carried out in [18] proves the remarkable reconstruction accuracy of the CS-TFM method in test waveforms as inspired by laboratory and real-world acquisitions. Therefore, the TFM stage is a plausible candidate to reproduce the functioning of a traditional modulation plus intermediate filter circuit and guarantees good flexibility when coupled with a CS-based support recovery.

### 3.3.4. Quasi-Peak Estimation

Quasi-peak detectors were originally conceived to evaluate the emissions' attack, integration and decay time. In the context of supraharmonic measurements, the most

widely employed is the CISPR-16 one [20]. In its analog realization, the CISPR-16 quasi-peak detector consists of a cascade of an RC circuit and a critically damped meter. For the sake of completeness, Table 1 reports the main specifications of the CISPR-16 detector, namely the charging $\tau_c$ and discharging time constant $\tau_d$ of the RC circuit, as well as the damped meter time constant $\tau_m$. This corresponds to an intermediate filter $-6$ dB bandwidth of 200 Hz, in compliance with the resolution of the aforementioned support recovery stage. It is also worth noticing that such a detector requires the input signal to be supplied in stationary conditions for an overall observation interval not lower than 2 s. The final measurement result consists of the maximum QP value recorded during the entire observation interval. It should be underlined that this peculiar measurement approach is intended not to simply evaluate the peak value of a given pulsated emission, but to account for the impact of repeated peaks in the emissions of radiated and conducted transmission systems.

**Table 1.** Specifications of the CISPR-16 QP detector in the supraharmonic range (Band A [20]).

| Parameter | Time (ms) |
|:---:|:---:|
| $\tau_c$ | 45 |
| $\tau_d$ | 500 |
| $\tau_m$ | 160 |

In recent years, digital realization of the CISPR-16 QP detector has been proposed in order to facilitate the integration within modern digital instrumentation and expedite the QP value estimation of simultaneous multi-tone emissions [41]. In this paper, a digital detector is also implemented following the modeling approach proposed in [42]. As thoroughly discussed in [33], digital and analog realizations are nearly coincident, with a discrepancy lower than 1.5 dB, given that the two input signals are also coincident (i.e., neglecting the distortion introduced by the analog front-end of the acquisition stage).

Based on the inherent functioning of the detector circuit, though, it makes no sense to compute the QP value on single segments of 0.5 ms. Once a supraharmonic component in a segment is identified, its time-domain profile is reconstructed and consecutive segments are merged together until an overall duration of 20 ms is achieved. Then, the QP detector is applied to the 20 ms reconstruction and the result is stored into a buffer. As a moving average, this approach is repeated at each new segment, and at the end of the measurement procedure, the final QP value is defined as the maximum of the values stored in the buffer. For the sake of consistency with most analog realizations of the CISPR-16 method, the buffer is designed as a First-In-First-Out queue whose length corresponds to the number of segments contained in 2 s.

### 3.3.5. Summary

The combination of the CS-TFM with a digital QP detector represents the actual novelty of this research and allows for completing the design of a measurement method for the metrological characterization of pulsating supraharmonic emissions. The CS-TFM part guarantees the robust and accurate identification of the supraharmonic components in terms of estimated parameters, namely instantaneous amplitude, initial phase, frequency and rate of change of frequency, but also in terms of the reconstructed time-domain profile. The QP detector part, instead, enables us to compare the measured values with the vast majority of traditional EMC meters that still apply the CISPR-16 representation.

The combination of the two routines has been deployed within the METAS PMU calibrator and is capable of associating with each measurement also a GPS-based time-stamp. In the case of QP values, the time-stamp corresponds to the time-stamp of the first segment that constitutes the 20 ms interval processed by the digital detector. In this way, it is possible to evaluate possible evolutions as a function of time, as well as to detect whether some portions of the signals were corrupted or badly recovered.

Similar measurements could be carried out by means of vector network analyzers or equivalent devices. Nevertheless, the objective of the SupraEMI project was to develop a new reference method whose computation time and complexity are in line with other power system monitoring applications, whereas existing commercial solutions typically operate offline.

## 4. Experimental Validation

### 4.1. Measurement Setup

The experimental validation relies on two different measurement setups that allow for reproducing both steady-state and pulsating emissions. For the sake of simplicity, the following tests consider only voltage signals, but equivalent setups and similar results can be obtained with current signals.

Figure 6 represents the block scheme of the measurement setup for the steady-state emissions. The PXI DAC board is responsible for generating the test waveforms, which consist of a fundamental component at 50 Hz and a single supraharmonic component whose frequency is varied between 9 and 150 kHz. As aforementioned, the DAC operates in continuous mode and therefore reproduces an ideal steady-state condition. In the following stage, the test waveform is amplified by means of a Rohrer PA 2021K Voltage Amplifier (Rohrer GmbH, Muenchen, Germany). Although the amplifier bandwidth does not exceed 30 kHz, its frequency response has been characterized in detail and the inherent low-pass filtering effect can be adequately compensated. By properly setting the DAC output range and the amplifier gain, it is thus possible to reproduce an operating condition where the fundamental component amplitude is equal to the nominal system value, i.e., 230 V, whereas the supraharmonic component amplitude is varied between 10, 5 and 1% of the fundamental one. In order to regulate the current output of the voltage amplifier, the signal passes through a calibrated load. Moreover, a programmable static impedance allows for mimicking the non-ideal effects of real-world circuitry. In particular, the programmable impedance has been developed in METAS workshop, and can be used either in purely resistive mode (10 $\Omega$) or in the presence of capacitive and inductive contributions (10 µF, 100 µH, 10 $\Omega$) [43]. Finally, the signal is supplied to a commercial electricity meter (EMH metering GmbH, Gallin, Germany) in order to reproduce a plausible measuring configuration as envisioned by the SupraEMI project.

As in any measurement related to power supply, an important aspect to address is the galvanic isolation. In this case, it is guaranteed by the voltage amplifier, whose current output is limited to 100 mA.

The same signal is measured in parallel by two different approaches. On one side, the signal is acquired by the PXI ADC via a TA 043 Differential Probe (Pico Technology, St Neots, UK), whose bandwidth and attenuation ratio are equal to 100 MHz and 1:100, respectively. The acquired waveform is then processed according to the method in Section 3. It is worth noticing that the ADC acquires also the voltage amplifier output: the comparison between the two acquired signals allows for assessing the distortion introduced by the static impedance, the calibrated load and the electricity meter. On the other side, the signal is first suitably attenuated and then supplied to a FPL1007 Spectrum Analyzer (Rohde & Schwarz GmbH, Munich, Germany).

For this analysis, we consider an overall test duration of 10 s and we compare the maximum QP value as provided by the FPL1007 and the proposed method. The first one represents the reference standard as it is calibrated against the CISPR-16 requirements, whereas the second one is evaluated as a possible candidate for laboratory measurements of supraharmonics in power systems.

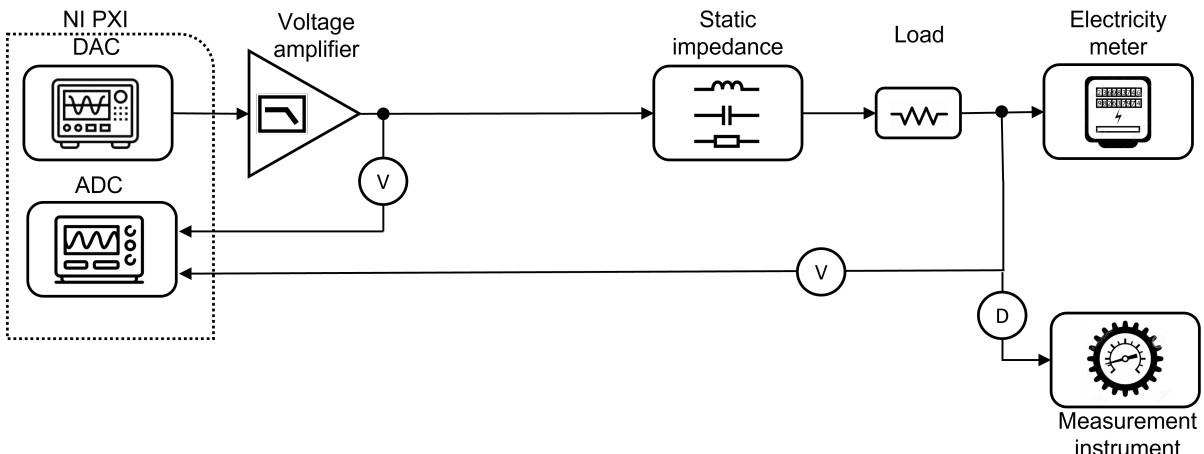

**Figure 6.** Measurement setup for the experimental validation in the presence of steady-state emissions.

In order to generate pulsating emissions, the measurement setup is slightly modified, as shown in Figure 7. In this case, the PXI DAC generates only the fundamental component at 50 Hz, whereas the synchronization board generates a trigger signal at a user-defined frequency (in the supraharmonic range) that drives an electronic IGBT switch. In this way, we are able to reproduce the effect of power electronic switching at high frequency in modern low-voltage appliances. Indeed, at the output of the switch, the signal consists of the fundamental component plus the harmonics of the switching frequency, whose amplitudes follow a decreasing trend dictated by the selected frequency and the switch circuitry.

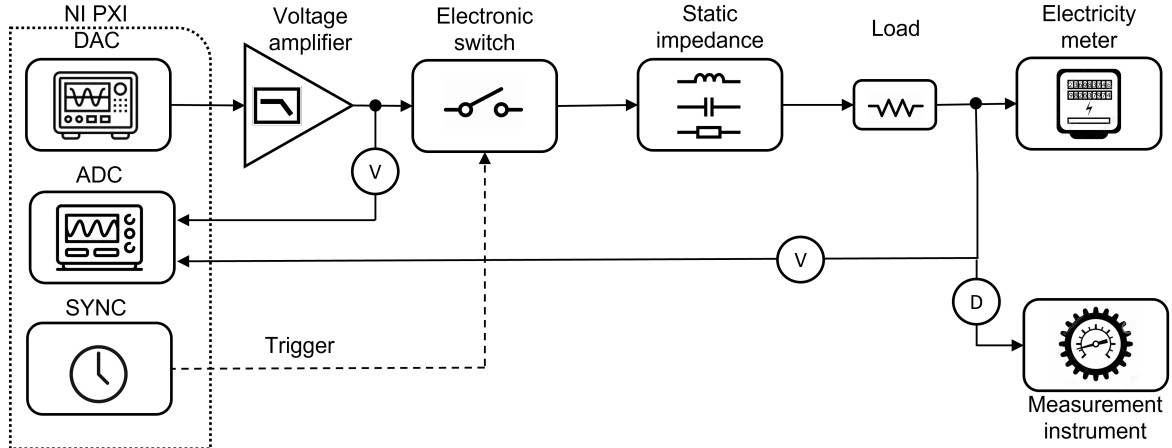

**Figure 7.** Measurement setup for the experimental validation in the presence of pulsating emissions.

### 4.2. Test Waveform Quality

Before proceeding with the QP value estimation, it is interesting to observe the test waveforms that can be generated by the considered measurement setups.

In this context, Figure 8 shows the time-domain representation of the test waveforms in three different configurations: steady-state emission with resistive impedance (black), and pulsating emission with resistive (blue) and complex impedance (red). For the sake of clarity, the fundamental component is superposed in yellow. In this regard, it is interesting to observe how the different configurations produce different distortion effects. In steady-state conditions, the supraharmonic interference is continuous, whereas in pulsating conditions, it is scaled to the fundamental magnitude, but only during the positive semiperiod (due to the specific functioning of the IGBT switch).

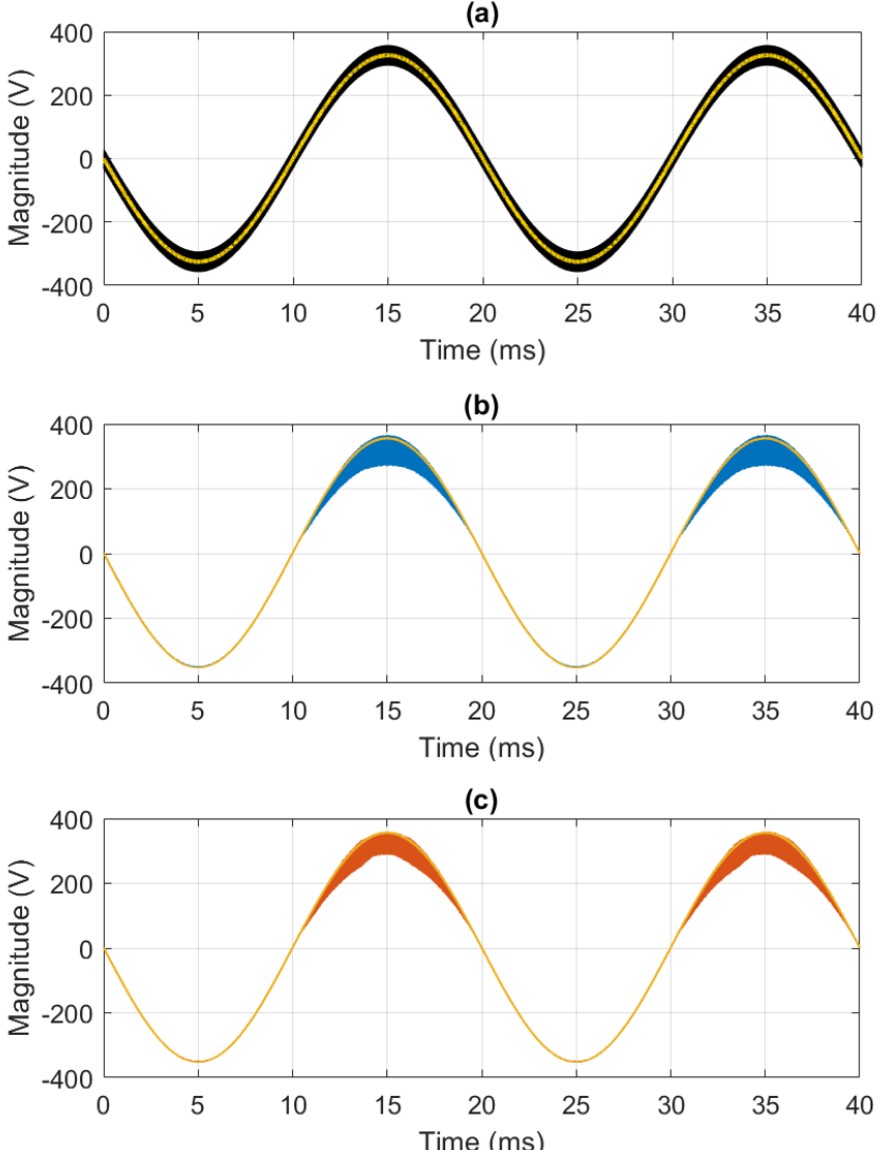

**Figure 8.** Time-domain representation of the acquired test waveforms in steady-state conditions with resistive impedance (**a**), and in pulsating conditions with resistive (**b**) and complex impedance (**c**). The yellow line indicates the fundamental component.

In Figure 9, instead, the same waveforms are presented in the frequency domain. In the upper plot, the entire range of interest is considered. It should be noticed how the voltage amplifier introduces some non-linearities even in the fundamental component only, whereas at higher frequencies, the switch introduces a dominant frequency (in this case, 25 kHz) and its harmonics (as in any square wave modulation). The lower plot focuses around the switching frequency and shows the effect of the different impedance configurations. In the purely resistive case, the switch modulation is characterized by a regular and symmetric decreasing trend, whereas the complex case exhibits a wider bandwidth with some discontinuities due to the non-linear effects of the inductive and capacitive components.

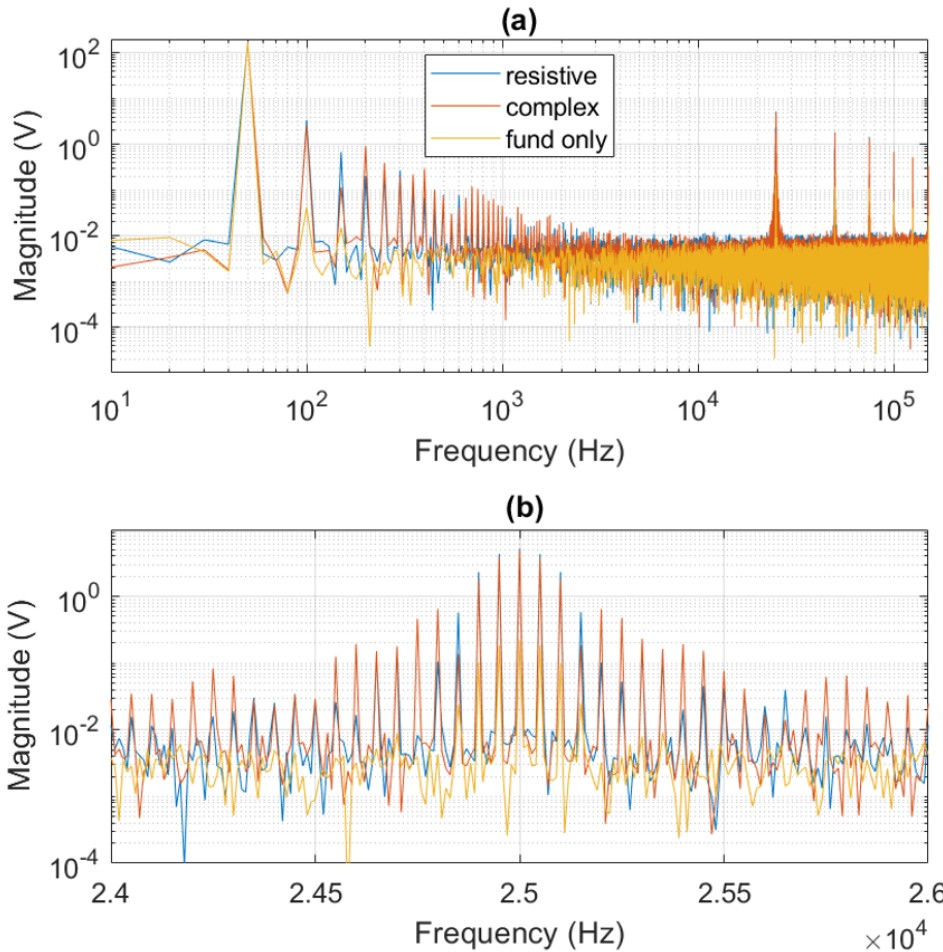

**Figure 9.** Frequency-domain representation (**a**) and detail of the first suprahamronic (**b**) of the acquired test waveforms in pulsating conditions with resistive (blue) and complex impedance (red). The yellow line indicates the fundamental component.

*4.3. QP Value Comparison*

First, the performance of the proposed method in the presence of steady-state emissions is evaluated. For this analysis, ten consecutive experiments with an overall duration of 10 s have been carried out. At the end of each experiment, only the maximum values provided by the reference instrument and by the proposed method are compared. In the following, the statistical distribution of the estimation errors is characterized in terms of mean, standard deviation and maximum. In this regard, Table 2 reports the selected metrics as a function of the supraharmonic component frequency and magnitude. For this analysis, the frequency varies between 25 and 150 kHz in order to span the entire supraharmonic range, whilst the magnitude ranges from 10 to 1 % of the fundamental component magnitude.

Based on the obtained results, the proposed method proves to be consistent with the reference instrument, with a worst-case deviation that is lower than 15 mV. Moreover, its performance is independent of the supraharmonic frequency, i.e., it guarantees the same level of accuracy over the entire supraharmonic range.

**Table 2.** Steady-state emissions—QP value estimation errors.

| Frequency (kHz) | Magnitude (V) | Mean (mV) | Std Dev (mV) | Max (mV) |
|:---:|:---:|:---:|:---:|:---:|
| | 23.0 | −13.495 | 0.805 | −14.810 |
| 25 | 11.5 | −8.310 | 0.441 | −9.053 |
| | 2.3 | −3.450 | 0.266 | −3.906 |
| | 23.0 | −13.569 | 0.678 | −14.365 |
| 50 | 11.5 | −8.438 | 0.302 | −8.857 |
| | 2.3 | −3.483 | 0.274 | − 3.838 |
| | 23.0 | −13.655 | 0.713 | −14.934 |
| 100 | 11.5 | −8.388 | 0.384 | −8.925 |
| | 2.3 | 3.484 | 0.200 | −3.908 |
| | 23.0 | −13.671 | 0.682 | −14.368 |
| 150 | 11.5 | −8.765 | 0.286 | −9.327 |
| | 2.3 | −3.418 | 0.220 | −3.738 |

A similar test has been carried out in the case of pulsating emissions. Figure 10 shows the recovered time-domain profile of the supraharmonic disturbances as provided by the proposed method. The adoption of a complex impedance introduces significant distortions that are nonetheless accurately tracked by the proposed method.

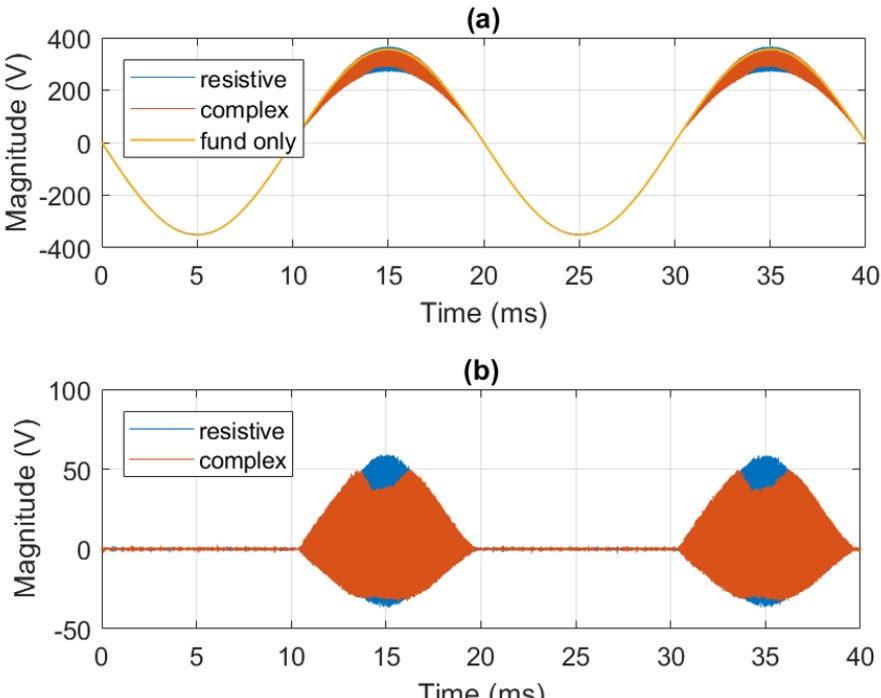

**Figure 10.** Time-domain profile of the acquired (**a**) and reconstructed signal (**b**) as provided by the proposed method.

In this context, Table 3 compares the QP values as provided by the proposed method and the reference instrument as a function of the switching frequency and of the impedance configuration. Once more, it is noticeable a significant correlation between the two sets of estimates. Moreover, the proposed method is capable of accurately capturing the QP reduction in the presence of inductive and capacitive effects.

**Table 3.** Steady-state emissions—QP value estimation errors.

| Frequency (kHz) | Impedance | Method (V) | Reference (V) |
|:---:|:---:|:---:|:---:|
| 25 | res | 15.017 | 15.040 |
|    | compl | 13.524 | 13.585 |
| 50 | res | 9.675 | 9.580 |
|    | compl | 8.005 | 8.385 |
| 100 | res | 5.127 | 5.033 |
|     | compl | 4.032 | 4.028 |
| 150 | res | 4.166 | 4.268 |
|     | compl | 3.144 | 3.050 |

*4.4. Uncertainty Budget*

By definition, a QP estimation involves an averaging process that tends to minimize the measurement noise. Nevertheless, based on the adopted setup, it is possible to draft a preliminary uncertainty budget.

As regards the reference value, the measurement chain consists of the DAC, the voltage amplifier, the signal attenuator and the spectrum analyzer. For this analysis, let us consider a test signal whose frequency and magnitude are set equal to 100 kHz and 23 V. The reported uncertainties refer to a cover factor of 2 and have been derived from the calibration certificate that is periodically issued by METAS laboratories. In terms of signal magnitude, in steady-state conditions, the generation stage (i.e., DAC plus amplifier) has an overall uncertainty of 38 ppm. By adding the attenuator and the spectrum analyzer, the overall uncertainty rises up to nearly 100 ppm.

As regards the proposed method, the measurement chain substitutes the signal attenuator and the spectrum analyzer with the differential voltage probe, the ADC and the algorithmic routine. The latter has a nearly negligible contribution, i.e., 20 ppm. In fact, the uncertainty budget is dominated by the voltage probe, whose uncertainty (after proper calibration) is equal to 500 ppm.

It should be noticed that the passive elements (i.e., static impedance, load, meter front-end) as well as the electronic switch have not been included in this preliminary budget as their contribution is several orders of magnitude lower than the voltage probe or attenuator. In the future steps of the research, a more rigorous and detailed assessment of uncertainty sources and propagation is envisioned.

As aforementioned, these uncertainty levels refer to single magnitude measurements, whereas the QP estimation involves a filtering and averaging procedure whose effect should be assessed via Monte Carlo simulations. Nevertheless, the remarkable accuracy of the two measurement setups confirms the reliability of the results presented in Section 4.3.

## 5. Conclusions

Due to the ever-increasing penetration of renewable energy sources and power electronics, modern smart grids are affected by high levels of harmonic distortion in the supraharmonic range, i.e., from 9 to 150 kHz. From a metrological point of view, an acknowledged reference method as well as a complete normative framework are still missing. In this paper, the problem of supraharmonic measurements in power system scenarios is taken into account. In particular, the possibility of combining a CS-TFM estimator for supraharmonic components with a digital QP detector model is discussed. The different stages of the algorithmic routine are described and the possible sources of uncertainties are briefly outlined and characterized via numerical simulations. Finally, the proposed method is experimentally validated against a commercial device, compliant with the CISPR-16 specifications and widely employed in electromagnetic calibration tests. The validation results confirm the potential of the proposed approach as a measurement method for supraharmonic conducted emissions in the field of modern smart grids.

**Funding:** This project (18NMR05 SupraEMI) has received funding from the EMPIR programme co-financed by the Participating States and from the European Union's Horizon 2020 research and innovation programme.

**Institutional Review Board Statement:** Not applicable.

**Informed Consent Statement:** Not applicable.

**Acknowledgments:** The author would like to thank Frederic Pythoud (METAS) for the insightful advice and for helping in the realization of the laboratory measurement campaigns.

**Conflicts of Interest:** The author declares no conflict of interest. The funders had no role in the design of the study; in the collection, analyses, or interpretation of data; in the writing of the manuscript, or in the decision to publish the results.

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
