# Peer review of "Measurement of Conducted Supraharmonic Emissions: Quasi-Peak Detection and Filter Bandwidth"

_2673-8244, doi:10.3390/metrology2020011_

Round 1
Reviewer 1 Report
quite impressive paper which adresses current power system operation challenges
Author Response
The Author thanks the Reviewer for her/his positive comment.
Reviewer 2 Report
In this paper, the author points out the absence of a normative framework for the quantification of emissions in the supraharmonic range, and proposed an implementation of an algorithmic approach to estimate traditional Quasi-Peak values. Its key argument is that a fully digital approach combining CS-TFM estimator with a digital QP detector model would allow for more effective quantification of supraharmonic emissions in real world conditions.
I thank the author for a well written paper and would like to suggest the following few minor revisions before publication:
- This paper is well written, but it is quite long (15 pages). It would be beneficial if the author could try to shorten it a bit by simplifying the language in order to bring out the essence of the argument.
- Line 26: Please add reference with weblink to SupraEMI webage.
- Line 43-45: Please add reference supporting argument that THD and SFDR not adequate at the discussed frequencies. Or expand discussion.
- Line 171, grammar: "have not yet been characterized"
- Line 174: please add reference to SpuraEMI reserach plan
- Figure 1: the symbols used for time-base, generation and acquisition are very similar and difficult to distinguish if printed on A4. I suggest to redraw this figure, use bigger symbols and find a way to differentiate them more. There is no legend for the various dashed and dotted lines.
- Line 209 : I would like to suggest that the author publish the code used (as Supplementary Information, or on github, or in any other form he deems suitable), see : https://www.nature.com/articles/467753a
- Figure 6 and 7: please clean up labels, avoid line return in word (Measuremen t). It is also difficult to read at A4 print size, maybe redraw it with bigger symbols
- line 473, typo: "selected"
Author Response
In this paper, the author points out the absence of a normative framework for the quantification of emissions in the supraharmonic range and proposed an implementation of an algorithmic approach to estimate traditional Quasi-Peak values. Its key argument is that a fully digital approach combining CS-TFM estimator with a digital QP detector model would allow for more effective quantification of supraharmonic emissions in real world conditions.
I thank the author for a well written paper and would like to suggest the following few minor revisions before publication:
1. This paper is well written, but it is quite long (15 pages). It would be beneficial if the author could try to shorten it a bit by simplifying the language in order to bring out the essence of the argument.
The Author would like to thank the Reviewer for her/his positive comment. As correctly pointed out by the Reviewer, the article is quite lengthy, as it provides many details that may result just auxiliary with respect to the main topic that is the validation of a new method for QP estimation of supraharmonic emissions. In the revised manuscript, the text has been shortened by removing repetitions and explanations of concepts that are quite basic in electrical metrology. Unfortunately, though, the additions required by other Reviewers have made practically impossible to reduce the manuscript length, but the Author hopes the readability and clarity have improved.
2. Line 26: Please add reference with weblink to SupraEMI webpage.
Noted. The link has been added in the references section. Thank you for the useful advice.
3. Line 43-45: Please add reference supporting argument that THD and SFDR not adequate at the discussed frequencies. Or expand discussion.
The Reviewer is right. The sentence was unclear. The intention was not to undermine the significance of THD and SFDR metrics, rather than to point out that in the considered rage of frequencies these metrics are not frequently reported, as they are cumulative indices and not related to a single component. The paragraph has been revised in order to better clarify this aspect.
4. Line 171, grammar: "have not yet been characterized"
Thank you for the careful reading. The manuscript has been proofread in order to detect other possible grammar errors and typos.
5. Line 174: please add reference to SpuraEMI research plan
Noted. The link has been added in the references section. Thank you for the useful advice.
6. Figure 1: the symbols used for time-base, generation and acquisition are very similar and difficult to distinguish if printed on A4. I suggest redrawing this figure, use bigger symbols and find a way to differentiate them more. There is no legend for the various dashed and dotted lines.
The Reviewer is right. The resolution of the different symbols is quite poor and thus difficult to distinguish in the printed version. The figure has been redrawn and the legend has been updated. The Author hopes it is now more readable and understandable.
7. Line 209: I would like to suggest that the author publish the code used (as Supplementary Information, or on github, or in any other form he deems suitable), see : https://www.nature.com/articles/467753a
Thank you for the useful suggestion! Part of the code is currently considered for a license and thus cannot be published until a final decision has been taken. But I will for sure upload a github with at least an executable version.
8. Figure 6 and 7: please clean up labels, avoid line return in word (Measuremen t). It is also difficult to read at A4 print size, maybe redraw it with bigger symbols
The Reviewer is right. This problem occurred in the conversion from png to vectorial format. The issue has been fixed in the revised version.
9. line 473, typo: "selected"
Thank you for the careful reading. The manuscript has been proofread in order to detect other possible grammar errors and typos.
Reviewer 3 Report
Interesting paper to measurement of conducted supraharmonic emissions. I have the following recommendations for the work:
- In measurements related to the power supply, one of the main principles is galvanic isolation. It can be explained how it was realized. Maybe through Omicron's transformers.
- Omicron offers a BODE 100 analyzer. It can be explained why it is not applicable in this case and the proposed concept is better.
In conclusion, Interesting research and I propose that it be published.
Author Response
Interesting paper to measurement of conducted supraharmonic emissions.
The Author would like to thank the Reviewer for her/his positive comment.
I have the following recommendations for the work:
• In measurements related to the power supply, one of the main principles is galvanic isolation. It can be explained how it was realized. Maybe through Omicron's transformers.
The Reviewer is right. The galvanic isolation is guaranteed by the voltage amplifier. This important aspect has been clarified in the revised version. Thank you very much for pointing out an aspect that could be misleading and highlighting a peculiarity of the measurement setup.
• Omicron offers a BODE 100 analyzer. It can be explained why it is not applicable in this case and the proposed concept is better.
Once more, the reviewer is absolutely right. There exist commercial solutions, e.g. the BODE 100 or other VNAs, that allow for measuring conducted emissions also in the supraharmonic range. Nevertheless, the objective of the SupraEMI project was to develop a new reference method whose computation time and
complexity was in line with other power system monitoring applications. In this context, the present paper summarizes a previously proposed method and verifies that its signal reconstruction does not affect the estimation of the QP value and thus can be compared to commercial VNAs, as suggested by the Reviewer.
The Author would like to thank the Reviewer for pointing out this aspect that is important to understand the actual objective of the research program.
In conclusion, interesting research and I propose that it be published.
The Author would like to thank the Reviewer for her/his positive comment.
Reviewer 4 Report
The manuscript presents a novel method for estimating the supraharmonic components in power system signals, based on previously developed algorithms. It is very well written, with few spelling and grammar issues, indicated in yellow in the attached PDF. The methods are well explained, with proper references to more detailed explanations, and the synthetic and experimental results are sound.
There only content issue is in line 206, as the unit Vrms does not exist. The only SI unit is the volt (V). It should be expressed as "300 V RMS".
Also, I believe the manuscript would benefit from a measurement uncertainty estimation, especially as it it being submitted to a metrology journal.

Author Response
The manuscript presents a novel method for estimating the supraharmonic components in power system signals, based on previously developed algorithms. It is very well written, with few spelling and grammar issues, indicated in yellow in the attached PDF. The methods are well explained, with proper references to more detailed explanations, and the synthetic and experimental results are sound.
The Author would like to thank the Reviewer for her/his positive comment.
There only content issue is in line 206, as the unit Vrms does not exist. The only SI unit is the volt (V). It should be expressed as "300 V RMS".
Thank you for the careful reading. The manuscript has been proofread in order to detect other possible grammar errors and typos.
Also, I believe the manuscript would benefit from a measurement uncertainty estimation, especially as it is being submitted to a metrology journal.
The Author would like to thank the Reviewer for her/his insightful and useful suggestion. A preliminary uncertainty budget has been added at the end of Section 4. Thank you!